# AME: Aligned Manifold Entropy for Robust unsupervised Vision-Language Distillation

## Abstract

Unsupervised knowledge distillation is a long-established technique for knowledge transfer, and has regained attention in the emergence of pre-trained vision-language models (VLMs). However, its objectives, i.e, probability distributions, are inherently scalar and directionless, which represent a sharp contrast to the similarity-based objectives employed in vision–language model training. As a result, the unsupervised distillation paradigm fails to impose sufficient cross-modal alignment, such alignment is essential for generalization in vision–language knowledge distillation. To address this major challenge arising from the representation misalignment, we propose **A**ligned **M**anifold **E**ntropy (AME) for robust unsupervised vision-language distillation (AME), aiming to achieve robust generalization in vision-language distillation tasks. Specifically, AME performs entropy compression over a restructured shared manifold (RSM), where multi-modal inputs (images and texts) are jointly embedded through projection functions. Here, we embed the features into a compact structure through representational compression, which in turn enforces directional alignment within the representation space. Note that AME keep the original backbone architecture without the need for additional modules. Thus, the proposed AME establishes a paradigm that effectively reinstates directional alignment and significantly improve representation convergence in low-data regimes. Extensive experiments and theoretical analysis across a wide range of settings demonstrate that AME is consistently conducive to robust unsupervised knowledge distillation, resulting in superior generalization across 11 datasets. Clearly, AME is a principled paradigm for unsupervised vision-language distillation, which advances it into a broader range of downstream tasks.

## 1 Introduction

Unsupervised knowledge distillation (UKD) is a widely adopted paradigm for model compression and knowledge transfer, where a compact student model is trained to approximate the predictive output of a larger teacher model by minimizing divergences between probability distributions. Although UKD has demonstrated strong effectiveness in downstream single-modality tasks, its extension to multi-modal settings is fundamentally non-trivial due to the misalignment across modalities. In particular, we argue that such misalignment in vision-language UKD originates from the reliance of conventional distillation theory on scalar-valued probability distributions produced by the softmax operation, which inherently keep no directional information critical for cross-modal alignment.

In contrast, a well-known success in cross-modal representation alignment is offered by CLIP (Radford et al., 2021) and ALIGN (Jia et al., 2021), which exploits the cosine similarity as training objective that simultaneously encodes both distance in the embedding space and directional relations between paired representations. This fundamental discrepancy explains why vision-language UKD is particularly prone to representation misalignment and, in turn, motivates the design of target distributions that explicitly embed similarity-driven directional cues to enforce consistent cross-modal alignment. With such effective cross-modal alignment, the representation obtains well-structured geometric properties (data geometric), which, as defined in Phuong & Lampert (2019), is consequently conducive to robust unsupervised knowledge distillation and clearer class separation.

Recent efforts attempt to extend unsupervised knowledge distillation into multi-modal tasks, exemplified by PromptKD (Li et al., 2024) and KDPL (Mistretta et al., 2024), which leverage learnable

prompts to reduce divergence from the teacher model in downstream tasks. While KDPL yields only marginal improvement in generalization, PromptKD typically requires large-scale training data to achieve any noticeable improvement, PromptKD fails to effectively align ambiguous or boundary-adjacent representations in low-data regimes, as illustrated in Figure 1(a). This observation suggests that its gains are the result of data compensation and circumventing the core misalignment challenge. As a result, despite incorporating either incremental architectural modifications or external knowledge, these approaches remain fundamentally constrained by conventional distillation theory, thereby lacking the directional objectives necessary for more robust cross-modal alignment.

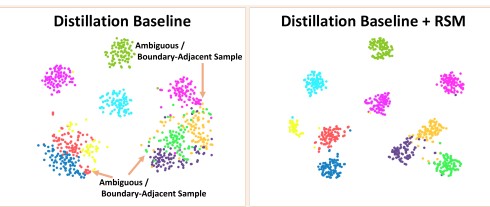 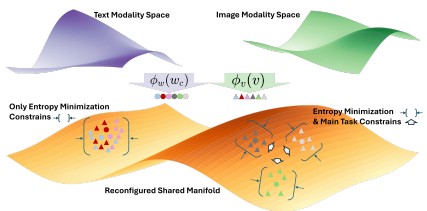

(a) OxfordPets visualization under 16-shot setting.  (b) Directional alignment under AME.

Figure 1. (a) Overview of representation quality and performance stability on OxfordPets under 16-shot settings, highlighting inter-class boundaries and representation separability across methods. (b) Illustration of the proposed Aligned Manifold Entropy (AME) mechanism, where entropy compression facilitates representation alignment in the unsupervised distillation process.

As this challenge arises from the coupling between representation misalignment and UKD, it is paramount to find a principle way to constrain cross-modal representation within a compact data geometry. Therefore in such distillation training, we leverage entropy compression which, similar to UKD, is an information-theory based approach. Accordingly, from the perspective of information theory, UKD ensures distributional alignment and entropy compression constrains representational entropy. Thus, such integration establishes a coherent and synergistic framework for vision-language distillation. However, we find that directly compressing the entropy in the multi-modal representation space fails to guarantee semantic cohesion, which in turn results in representation misalignment and high-entropy representations, rendering to distillation ineffectively in multi-modal tasks.

To effectively leverage entropy compression, we are inspired by the multiple signal combination philosophy in Maximum Ratio Combining (MRC) (Simon & Alouini, 2004) and the low-dimensional representational method in the Manifold Hypothesis (Tenenbaum et al., 2000; Roweis & Saul, 2000), and develop a novel paradigm to embed multi-modal features into a common latent space with low entropy. In MRC, signals received from multiple antennas are coherently aggregated by multiplying each with the complex conjugate of its corresponding channel coefficient prior to summation. Accordingly, in our case, we aggregate multi-modal representations to construct a shared latent space in which representations are coherently aligned and geometrically paired. Here, this shared representation space can be viewed as the aggregated signal system, whose entropy inevitably introduces redundant information during the aggregation process. Following the mechanism behind manifold hypothesis, we project these representations onto a low-dimensional manifold to eliminate redundant entropy. Explicit entropy compression of the reconfigured shared manifold (RSM) then drives multi-modal representations to converge toward a common underlying structure.

Ultimately, the synergy between entropy compression in RSM and the distillation objective prevents the collapse of text and image embeddings into trivial clusters. Instead, it drives a geometrically consistent alignment, thereby yielding clearer class separation, as illustrated in Figure 1(b). Note that the proposed RSM is a lightweight, theoretically grounded module that can be integrated seamlessly into standard knowledge distillation pipelines. Importantly, RSM is model-agnostic and requires no architectural modifications, thereby ensuring a broader range of applications. As a result, unsupervised knowledge distillation with RSM enables the model to effectively generalize ambiguous or boundary-adjacent samples, significantly benefiting class separation. We refer to this paradigm as **A**ligned **M**anifold **E**ntropy (AME) for Robust Unsupervised Vision–Language Distillation. In essence, AME transforms the learning process from a scalar-level distributional matching objective into a principled framework that incorporates directional alignment. Consequently, it leads to significant improvement in UKD performance under low-data regimes for multi-modal tasks.

In summary, this paper makes the following major contributions:

- We propose a distillation paradigm, namely Aligned Manifold Entropy for Robust unsupervised Vision-Language Distillation (AME), which integrates entropy compression over a reconfigured shared manifold of multi-modal features to reinstate the directional nature of the training objective, thereby promoting principled representation alignment and enabling superior generalization under low-data regimes for multi-modal downstream tasks.

- We propose Reconfigured Shared Manifold compression (RSM), a lightweight yet theoretically principled module that can be seamlessly integrated into existing unsupervised vision-language knowledge distillation methods, enabling robust and effective distillation.

- We provide a theoretical account showing that the interplay between distillation and entropy minimization enforces robust representation alignment in vision–language models, establishing the theoretical foundation of our proposed paradigm, AME.

- In extensive experiments, our proposed AME achieves significant improvements over the distillation baseline in Base-to-New generalization and Cross-Dataset generalization, with notable performance gains of at least 2.79% and 1.92%, respectively.

## 2 LITERATURE REVIEW

### 2.1 VISION-LANGUAGE MODELS

Vision language models (VLMs) have made significant advances in learning multi-modal representations through training on large-scale datasets. These representations exhibit strong generalization across a variety of downstream tasks, including image recognition (Gao et al., 2021a; Kim et al., 2022; Zhang et al., 2021), object detection (Feng et al., 2022; Maaz et al., 2022; Zang et al., 2022; Wang et al., 2019), and segmentation (Ding et al., 2022; Li et al., 2022; Rao et al., 2022). Building on this foundation, several approaches, such as BAN (Kim et al., 2018), Intra-Inter (Gao et al., 2021b), and MCAN (Yu et al., 2019), improve task performance by leveraging attention-based architectures. Alternatively, models like ViLBERT (Lu et al., 2019), LXMERT (Tan & Bansal, 2019), and UNITER (Chen et al., 2020) focus on vision-language learning through BERT-style architectures, achieving further performance gains. Among these approaches, CLIP (Radford et al., 2021), ALIGN (Jia et al., 2021), and LiT (Zhai et al., 2022) have catalyzed a shift toward a new paradigm that aligns images and texts via their respective encoders, leading to superior performance in downstream tasks. However, compared to fine-tuning methods (Dong et al., 2022), pretrained VLMs exhibit suboptimal performance on specific tasks. This has motivated a line of research aimed at effectively adapting vision-language models (VLMs) or transferring their knowledge to models tailored for specific downstream tasks (Gu et al., 2021; Lüddecke & Ecker, 2022; Wang et al., 2023).

### 2.2 KNOWLEDGE DISTILLATION

Knowledge distillation (KD) (Hinton et al., 2015) is a widely adopted technique for knowledge transfer, in which a lightweight model (student) is trained to match output logits or intermediate representations of a larger pretrained model (teacher). Consequently, the student model can achieve improved task-specific performance, such as image classification (Beyer et al., 2022; Chen et al., 2019; Hinton et al., 2015) and image segmentation (Dou et al., 2020; He et al., 2019), while maintaining a compact architectural design. Among the proposed KD methods, DML (Zhang et al., 2018) introduces a strategy to train the student and teacher models simultaneously, thereby improving distillation performance without relying on a powerful teacher network. Alternatively, DKD (Zhao et al., 2022) decouples the distillation loss into two components, target-class KD and non-target-class KD, to better characterize the difficulty of training samples and explain the effectiveness of logit-based distillation. PromptKD (Li et al., 2024) extends knowledge distillation to the prompt tuning paradigm, demonstrating superior generalization across a variety of downstream tasks. More recently, KDPL (Mistretta et al., 2024) improves distillation efficiency by updating only a limited number of parameters (i.e., the prompt), making the process more parameter-efficient. In contrast to existing baselines (Yun et al., 2020; Yuan et al., 2020; Zhang et al., 2018; Zhao et al., 2022; Li et al., 2024; Mistretta et al., 2024), we propose an entropy-based method that regularizes the output of the student model, aiming to enhance knowledge distillation performance under low-data regimes.

## 3 METHODOLOGY

We begin by revisiting the existing framework underlying our design, after which we introduce the architecture of the proposed AME. We then provide a theoretical analysis, based on entropy-theoretic principles, which elucidates how our proposed AME reinstates the directional nature of training objectives. In line with this analysis, feature-embedding visualizations, i.e, t-SNE and class-wise feature importance, exhibit structural regularities that corroborate our theoretical claims.

### 3.1 EXISTING BASELINE FRAMEWORK: VISION LANGUAGE MODELS

Existing vision language models, such as CLIP (Radford et al., 2021), are trained on large-scale image-text pair data, leaping forward the generalization capability across a wide range of multi-modal downstream tasks. Note that in CLIP, the image and text encoders transform the respective visual inputs and textual descriptions into embeddings, which are subsequently aligned using a contrastive loss. Specifically, the textual description is generated using the templates, e.g., "a photo of a {Class}", where $Class : C \in \{1, 2, ...C\}$. Such descriptions are then fed into the text encoder $\mathcal{F}_w$, resulting in text embeddings $W = \{w_i\}_{i=1}^C$. The image embeddings $V = \{v_j\}_{j=1}^N$ are similarly generated by its encoder $\mathcal{F}_v$, where $N$ is the batch size. Finally, the predicted label $\hat{y}$ corresponds to the text embedding that achieves the highest cosine similarity $sim(\cdot)$ with the given image embedding, which can be formulated as

$$p(\hat{y}|v) = \frac{\exp\big(\text{sim}(v, w_{\hat{y}})/\tau\big)}{\sum_{i=1}^{C} \exp\big(\text{sim}(v, w_i)/\tau\big)}, \tag{1}$$

where $\tau$ denotes a temperature parameter.

To unlock the potential of CLIP, a new paradigm, prompt learning has emerged to address the limitations of hand-crafted textual descriptions, which hinder the broader adoption of CLIP across downstream tasks. In such approaches, learnable tokens $P$ are prepended to the vectorized image and text inputs, which are then processed by the corresponding encoders, formally expressed as $V^p = \mathcal{F}_v([P^v, V])$ for the image branch and $W^p = \mathcal{F}_w([P^w, W])$ for the text branch, where $P^v$ and $P^w$ denote the visual and textual prompts, respectively. Accordingly, the prediction obtained with the learnable tokens is redefined as

$$p(\hat{y}|v^p) = \frac{\exp\big(\text{sim}(v^p, w_{\hat{y}}^p)/\tau\big)}{\sum_{i=1}^{C} \exp\big(\text{sim}(v^p, w_i^p)/\tau\big)}. \tag{2}$$

### 3.2 EXISTING BASELINE FRAMEWORK: UNSUPERVISED KNOWLEDGE DISTILLATION (UKD).

In UKD, the student model replicates the predictive behavior of a well-trained teacher model by minimizing the discrepancy between their output distributions. Here, the Kullback-Leibler (KL) divergence loss is commonly used to calculate the discrepancy, and is defined as

$$\mathcal{L}_{KD} = \tau^2 D_{KL}(\sigma(z_t/\tau) \parallel \sigma(z_s/\tau)). \tag{3}$$

Here, $\sigma$ denotes the softmax function and $\tau$ is the temperature parameter, while $z_t$ and $z_s$ represent the output distributions of the well-trained teacher model and student model, respectively.

### 3.3 ALIGNED MANIFOLD ENTROPY FOR ROBUST UNSUPERVISED VISION-LANGUAGE DISTILLATION

Building on the aforementioned insights into existing challenges, we propose Aligned Manifold Entropy for Robust Vision-Language Distillation (AME), a paradigm for vision-language knowledge distillation that reinstates the inherently directional nature of cross-modal alignment. The major architecture of AME is shown in Fig. 2. Specifically, the image and text embeddings, processed by their respective encoders, are then projected via a low-dimensional convolution-based function $V' = \phi_v(V)$ and a low-dimension MLP-based function $W' = \phi_w(W)$. These resulting representations are used exclusively within the reconfigured shared manifold. Subsequently, the reconfigured

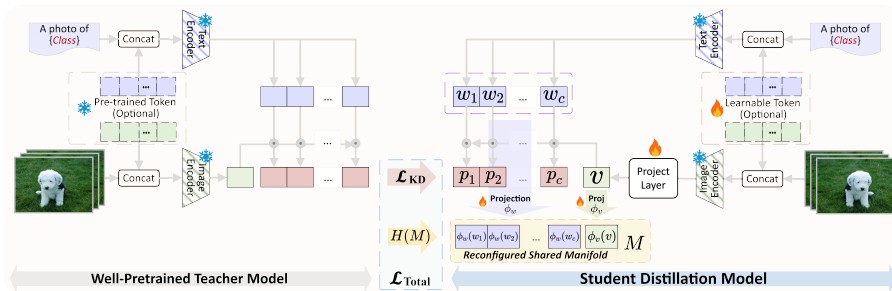

Figure 2. Overview of Aligned Manifold Entropy for Robust Unsupervised Vision-Language Distillation (AME). Given a well-pretrained teacher model and a lightweight student architecture, AME reconfigures a shared manifold for cross-modal feature alignment via entropy minimization.

latent space $M \in \mathbb{R}^{(N+C) \times R}$ is formed by row-wise concatenation of the projected text embeddings and image embeddings, such that $M = [\phi_w(w_{i=1}), \ldots, \phi_w(w_{i=C}), \phi_v(v_{j=1}), \ldots, \phi_v(v_{j=N})]^\top$.

Here, the RSM contributes by constructing a low-entropy shared multi-modal space that suppresses redundant entropy and condenses only salient cross-modal mutual information, thereby benefiting the data geometry and clarifying the convergence trajectory of representation alignment. The results are then averaged along the feature dimension to yield aggregated scalar activations, defined as

$$s_t = \frac{1}{R} \sum_{r=1}^{R} M_t \quad , \quad t = 1, \ldots, N+C. \tag{4}$$

Applying a softmax to the scalar scores $s_t$ produces normalized probabilities $p_t$, which characterize the entropic weight of each embedding in the reconfigured shared manifold $M$ that jointly integrates both text and image representations. The information entropy of $M$ is then formalized as

$$H(M) = -\sum_{t}^{N+C} p_t \log p_t, \tag{5}$$

Thus, the total loss function that integrates information entropy minimization over $M$ is given by

$$\mathcal{L}_{\text{Total}} = \mathcal{L}_{KD} + \omega H(M), \tag{6}$$

where $\omega$ is a weighting coefficient. Note that, the pseudo code of AME is provided in the appendix.

With this new training paradigm, distillation obtains directional alignment akin to the cosine similarity in VLMs, while the synergy between distillation and entropy compression further enforces effective and robust representation alignment within training process. It is worth noting that our paradigm is firmly rooted in information theory and does not introduce any additional embeddings, making it naturally compatible with a broad range of vision-language distillation frameworks.

### 3.4 THEORETICAL ANALYSIS

To elucidate the underlying mechanism of AME, we develop a theoretical analysis through the lenses of information theory and generalization theory. In particular, this section formalizes the roles of the reconfigured shared manifold and entropy minimization within the proposed distillation paradigm. Further theoretical analysis, covering the upper bound of the generalization error, the effect of sample size, and the complete proofs of all results, are deferred to the Appendix.

By characterizing the interactions between the RSM and the optimization objective, we aim to explore the mechanism through which the proposed method enhances performance. Under typical optimization dynamics, minimizing $H(M)$ encourages each probability $p_t$ to obtain a one-hot distribution. This implies the corresponding score $s_t$ must be maximized at a specific index to effectively reduce entropy. Accordingly, the gradient of the loss with respect to $s_t$ can be expressed as:

$$\frac{\partial H(M)}{\partial p_t} = -\sum_{t}^{N+C} (1 + \log p_t). \tag{7}$$

This gradient behavior encourages all $p_t$ values to converge toward the dominant score $p_{t*}$, leading the corresponding feature vectors $M_{t,:}$ to collapse toward a single point. As a result, the logits in $\mathcal{L}_{KD}$ become degenerate, ultimately undermining the class separability necessary for training.

### 3.4.1 IMPLICIT CLASS-WISE ALIGNMENT IN RECONFIGURED SHARED MANIFOLD

**theorem 1.** *Given the task constraint imposed by $\mathcal{L}_{KD}$, jointly minimizing the total loss $\mathcal{L}_{Total}$ requires the optimal solution to achieve intra-class concentration while preserving inter-class separability. At convergence, the optimization ensures that for any text embedding $w_c$ and its corresponding set of image embeddings $\{v_j : label(j) = c\}$ belonging to class $c$, the text and image representations in the reconfigured shared manifold $M$ satisfy the following condition:*

$$\phi_w(w_c) = \phi_v(v_j) = h_c, \quad \forall j : label(j) = c, \tag{8}$$

$$\|h_c - h_{c'}\| \geq \zeta > 0, \quad \forall c \neq c'. \tag{9}$$

*Consequently, the embeddings belonging to the same class collapse toward a class-specific representation vector $h_c$, effectively reducing intra-class entropy, as illustrated in the lower right of Figure 1b. Meanwhile, embeddings from different classes maintain sufficient separation to satisfy the knowledge distillation constraint $\mathcal{L}_{KD}$, achieving structured compression of the representations.*

The theorem demonstrates that the KL divergence loss between the teacher and student models enforces probabilistic compression between text and image. This loss implicitly induces a conditional joint distribution $p(W, V \mid S)$ over the reconfigured shared manifold, where $S$ denotes the input sample (e.g., an image-text pair). As a result, the information entropy $H(M)$ can be interpreted as an approximation of the conditional joint entropy $H'(W, V \mid S)$.

### 3.4.2 INFLUENCE OF THE INFORMATION ENTROPY TERM ON MUTUAL INFORMATION.

**Corollary 1.** *By incorporating $H(M)$ into the loss function, minimizing $H(M)$, which approximates the conditional joint entropy $H(W, V \mid S)$, effectively increases the conditional mutual information $I(W; V \mid S)$ beyond what is achieved by the KL divergence term alone.*

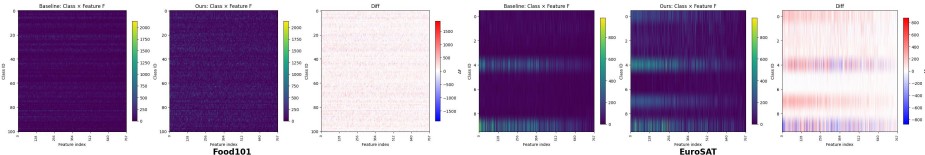

Figure 3. Class–feature relevance heatmaps on Food101 and EuroSAT. This illustrates the class–feature relevance heatmaps computed via one-vs-rest F-scores across representation dimensions, where each column corresponds to a dimension and each row to a class. Brighter intensities signify stronger discriminative utility of a dimension for identifying the target class.

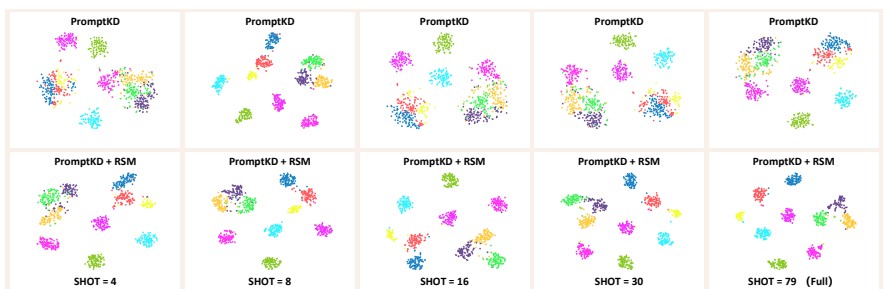

Figure 4. t-SNE visualization on the OxfordPets dataset across different shot settings.

This reveals a dual mechanism: probabilistic alignment via KL divergence and structural compression via entropy minimization, both of which jointly improve cross-modal representation learning.

As shown in Figure 3, relative to the baseline method (A), our proposed paradigm (B) yields consistently stronger and more widespread activation patterns across both dimensions and classes, indicating that a larger portion of the representation space is aligned with class-discriminative signals. For completeness, the class–feature relevance heatmaps of all 11 datasets are included in the appendix.

As a result, t-SNE visualization shown in Fig. 4 is conducted for both PromptKD and PromptKD + RSM under different training shot settings. It can be clearly seen that PromptKD + RSM exhibits more distinct discriminative clustering compared to the original PromptKD across all training shot settings on the OxfordPets dataset. Notably, well-defined class boundaries emerge persistently in PromptKD + RSM, while PromptKD often yields entangled and disordered structures under 4-shot, 16-shot, 30-shot, and full-shot scenarios. These results collectively demonstrate that our paradigm induces representations with significantly higher class separability and richer discriminative geometry, thereby establishing a more robust foundation for multi-model representation alignment.

# 4 EXPERIMENTS

AME is a theoretical framework requiring no architectural modifications to the backbone. It can therefore be deployed as a plug-and-play module, referred to as Reconfigured Shared Manifold compression (RSM), allowing seamless integration into a wide range of knowledge distillation frameworks for performance enhancement under data-scarce conditions. The backbone in Simple is frozen and a learnable projection function is introduced after the image encoder. Following the evaluation protocol of PromptKD, we compare these methods with well-established baselines across 11 datasets under two evaluations, i.e., Base-to-New Class and Cross-Dataset Generalization. In this section, the details of the experimental setup, including the training and evaluation protocols, baselines, are are firstly presented in sequence, followed by fundamental experiments. Note that dataset information, additional experiments, and further analyses are provided separately in the appendix.

Table 1. Comparison with existing approaches across 11 datasets and their averages. Here, RSM modules are appended to PromptKD (" PKD+ RSM"). While † indicates 16-shot training and ‡ indicates full-shot training, with the number of shots shown in parentheses after the dataset name.

### (a) Average

| Name | Base | New | HM |
|---|---|---|---|
| CLIP | 69.34 | 74.22 | 71.70 |
| CoOp | 82.69 | 63.22 | 71.66 |
| MaPLe | 82.28 | 75.14 | 78.55 |
| PromptSRC | 84.19 | 75.28 | 79.49 |
| PromptKD† | 80.36 | 74.64 | 77.40 |
| PromptKD‡ | 85.21 | 79.13 | 82.06 |
| PKD + RSM† | 83.31 | 77.30 | 80.19 |
| PKD + RSM‡ | 86.27 | 79.89 | 82.96 |

### (b) ImageNet ($\approx 1281$)

| Name | Base | New | HM |
|---|---|---|---|
| CLIP | 72.43 | 68.14 | 70.22 |
| CoOp | 76.47 | 67.88 | 71.92 |
| MaPLe | 76.66 | 70.54 | 73.47 |
| PromptSRC | 77.36 | 70.58 | 73.81 |
| PromptKD† | 77.02 | 71.03 | 73.90 |
| PromptKD‡ | 76.04 | 70.74 | 73.30 |
| PKD + RSM† | 76.03 | 70.21 | 73.00 |
| PKD + RSM‡ | 75.73 | 71.04 | 73.31 |

### (c) Caltech101 ($\approx 41$)

| Name | Base | New | HM |
|---|---|---|---|
| CLIP | 96.84 | 94.00 | 95.40 |
| CoOp | 98.00 | 89.81 | 93.73 |
| MaPLe | 97.74 | 94.36 | 96.02 |
| PromptSRC | 97.98 | 93.92 | 95.91 |
| PromptKD† | 98.34 | 95.23 | 96.76 |
| PromptKD‡ | 98.90 | 96.47 | 97.67 |
| PKD + RSM† | 98.56 | 95.67 | 97.09 |
| PKD + RSM‡ | 98.82 | 96.07 | 97.42 |

### (d) OxfordPets ($\approx 79$)

| Name | Base | New | HM |
|---|---|---|---|
| CLIP | 91.17 | 97.26 | 94.12 |
| CoOp | 93.67 | 95.29 | 94.47 |
| MaPLe | 95.43 | 97.76 | 96.58 |
| PromptSRC | 95.43 | 97.30 | 96.35 |
| PromptKD† | 90.13 | 92.54 | 91.32 |
| PromptKD‡ | 93.39 | 95.79 | 94.57 |
| PKD + RSM† | 95.23 | 97.07 | 96.14 |
| PKD + RSM‡ | 96.38 | 98.23 | 97.30 |

### (e) StanfordCars ($\approx 32$)

| Name | Base | New | HM |
|---|---|---|---|
| CLIP | 63.37 | 74.89 | 68.65 |
| CoOp | 78.12 | 60.40 | 68.13 |
| MaPLe | 72.94 | 74.00 | 73.47 |
| PromptSRC | 78.53 | 75.23 | 76.84 |
| PromptKD† | 79.48 | 81.40 | 80.43 |
| PromptKD‡ | 76.75 | 78.21 | 77.47 |
| PKD + RSM† | 80.30 | 81.54 | 80.92 |
| PKD + RSM‡ | 82.60 | 83.44 | 83.02 |

### (f) Flowers102 ($\approx 40$)

| Name | Base | New | HM |
|---|---|---|---|
| CLIP | 72.08 | 77.80 | 74.83 |
| CoOp | 97.60 | 59.67 | 74.06 |
| MaPLe | 95.92 | 72.46 | 82.56 |
| PromptSRC | 98.04 | 74.82 | 84.87 |
| PromptKD† | 95.38 | 77.28 | 85.38 |
| PromptKD‡ | 96.26 | 80.05 | 87.41 |
| PKD + RSM† | 98.58 | 81.65 | 89.32 |
| PKD + RSM‡ | 99.31 | 82.55 | 90.16 |

### (g) Food101 ($= 500$)

| Name | Base | New | HM |
|---|---|---|---|
| CLIP | 90.10 | 91.22 | 90.66 |
| CoOp | 88.33 | 82.26 | 85.19 |
| MaPLe | 90.71 | 92.05 | 91.38 |
| PromptSRC | 90.84 | 91.54 | 91.19 |
| PromptKD† | 78.56 | 81.29 | 79.90 |
| PromptKD‡ | 92.39 | 93.78 | 93.08 |
| PKD + RSM† | 89.48 | 91.06 | 90.26 |
| PKD + RSM‡ | 92.52 | 93.73 | 93.12 |

### (h) FGVCAircraft ($\approx 33$)

| Name | Base | New | HM |
|---|---|---|---|
| CLIP | 27.19 | 36.29 | 31.09 |
| CoOp | 40.44 | 22.30 | 28.75 |
| MaPLe | 37.44 | 35.61 | 36.50 |
| PromptSRC | 42.82 | 36.77 | 39.57 |
| PromptKD† | 44.42 | 39.81 | 41.99 |
| PromptKD‡ | 47.84 | 41.33 | 44.35 |
| PKD + RSM† | 45.32 | 39.25 | 42.07 |
| PKD + RSM‡ | 48.34 | 41.39 | 44.60 |

### (i) SUN397 ($\approx 40$)

| Name | Base | New | HM |
|---|---|---|---|
| CLIP | 69.36 | 75.35 | 72.23 |
| CoOp | 80.60 | 65.89 | 72.51 |
| MaPLe | 80.82 | 78.70 | 79.75 |
| PromptSRC | 82.59 | 78.71 | 80.60 |
| PromptKD† | 82.64 | 80.48 | 81.55 |
| PromptKD‡ | 83.48 | 81.38 | 82.42 |
| PKD + RSM† | 82.25 | 79.94 | 81.08 |
| PKD + RSM‡ | 83.56 | 81.30 | 82.41 |

### (j) DTD ($= 60$)

| Name | Base | New | HM |
|---|---|---|---|
| CLIP | 53.24 | 59.90 | 56.37 |
| CoOp | 79.44 | 41.18 | 54.24 |
| MaPLe | 80.36 | 59.18 | 68.16 |
| PromptSRC | 82.83 | 61.19 | 70.39 |
| PromptKD† | 75.42 | 60.51 | 67.15 |
| PromptKD‡ | 85.42 | 71.05 | 77.58 |
| PKD + RSM† | 84.15 | 69.16 | 75.92 |
| PKD + RSM‡ | 86.04 | 70.37 | 77.42 |

### (k) EuroSAT ($\approx 1350$)

| Name | Base | New | HM |
|---|---|---|---|
| CLIP | 56.48 | 64.05 | 60.03 |
| CoOp | 92.19 | 54.74 | 68.69 |
| MaPLe | 94.07 | 73.23 | 82.35 |
| PromptSRC | 93.35 | 69.29 | 79.54 |
| PromptKD† | 75.47 | 61.55 | 67.80 |
| PromptKD‡ | 97.32 | 80.27 | 87.98 |
| PKD + RSM† | 79.94 | 65.03 | 71.72 |
| PKD + RSM‡ | 97.60 | 81.19 | 88.64 |

### (l) UCF101 ($\approx 75$)

| Name | Base | New | HM |
|---|---|---|---|
| CLIP | 70.53 | 77.50 | 73.85 |
| CoOp | 84.69 | 56.05 | 67.46 |
| MaPLe | 83.00 | 78.66 | 80.77 |
| PromptSRC | 86.35 | 78.71 | 82.35 |
| PromptKD† | 87.12 | 79.90 | 83.36 |
| PromptKD‡ | 89.50 | 81.32 | 85.22 |
| PKD + RSM† | 86.54 | 79.66 | 82.96 |
| PKD + RSM‡ | 88.09 | 79.52 | 83.58 |

## 4.1 Experiments Setup

The implementation details adhere to the settings as PromptKD unless otherwise noted. Accordingly, all distillation methods in this paper use a well-trained PromptSRC (Khattak et al., 2023b) as teacher model, which is provided by PromptKD. For the training of the student model, CLIP with the ViT-B/16 backbone is used, while optimization is performed for 20 epochs with the stochastic gradient descent (SGD) optimizer, using a batch size of 8 and a learning rate of 0.005. The weighting coefficient $\omega$ in training loss is set to 50 in our case. The evaluation metric is defined as the average accuracy over three runs using random seeds 1, 2, and 3. For each run, accuracies are computed separately for base classes (Base), new classes (New), and their harmonic mean (HM). Note that in the full-shot setting, PromptKD, and model with RSM are trained on ImageNet for two epochs.

For the Baselines, we consider several well-known prompt tuning approaches as baselines, including CLIP (Radford et al., 2021), CoOp (Zhou et al., 2022b), MaPLe (Khattak et al., 2023a), PromptSRC (Khattak et al., 2023b), and PromptKD, which cover both single- and multi-modal paradigms.

## 4.2 Fundamental Experiments

### 4.2.1 Base-to-New Class Generalization.

Following Li et al. (2024), both the training and test datasets are split into base and new classes. The student model is trained on the unlabeled training set, while its performance is evaluated on the test set for both base and new classes. As shown in Table 1, we evaluate whether integrating our proposed RSM module, denoted as "+ RSM", into existing KD models improves knowledge transfer under both 16-shot and full-shot settings, by comparing with the baseline models.

Under the 16-shot training setting for distillation, it can be calculated in Table 1 that PromptSRC achieves the best average performance among all baselines across 11 datasets, achieving 84.19%, 75.28% and 79.49% for Base, New, and HM, respectively. Alternatively, PromptKD exhibits suboptimal performance, with corresponding results of 80.36%, 74.64% and 77.40%. Notably, PKD + RSM yields significant improvements over PromptKD by 2.94%, 2.66% and 2.79% on Base, New, and HM, respectively. It can be clearly seen that PKD + RSM surpass the best generalization provided by baselines on the New class and HM metrics, and achieve the best HM on 7 out of 11 datasets, including Caltech101, StanfordCars, Flowers102, FGVCAircraft, SUN397 DTD and UCF101. Note that on OxfordPets, Flowers102, Food101, DTD, and EuroSAT, PKD + RSM outperforms PromptKD in terms of HM, achieving 96.14%, 89.32%, 90.26%, 75.92% and 71.72%, respectively. These correspond to relative improvements of 4.82%, 3.94%, 10.36%, 8.78% and 3.92%. Such results indicate the superior effectiveness of proposed RSM in the low-data regime.

With the full-shot training setting, it can be seen in Table 1, PromptKD obtains the best average performance for all evaluation metrics ( 85.21%, 79.13% and 82.06% for Base, New, and respectively) among the baselines. By leveraging our proposed method, PKD + RSM achieves further improvement in overall performance, with gains of 1.06%, 0.77% and 0.90% on the Base, New, and HM, respectively. In terms of HM, improvements are observed on 7 out of 11 datasets including ImageNet, OxfordPets, StanfordCars, Flowers102, Food101, FGVCAircraft, and EuroSAT. Notably, RSM significantly improves the generalization performance of both PromptKD-based model across all metrics. For example, in terms of HM, the improvements from the PKD + RSM are 2.73%, 5.54%, and 2.75% on OxfordPets, StanfordCars, and Flowers102 datasets, respectively.

Overall, based on the comparison across different training setting in Table 1, our proposed RSM is shown to consistently enhance generalization performance in knowledge distillation. These results suggest that our proposed RSM serves as a universal and effective module, enabling robust generalization across a wide range of downstream tasks under the low-data regime.

### 4.2.2 Cross-Dataset Generalization

In Cross-Dataset generalization, the student model is trained on unlabeled images of unseen classes, similar to the Base-to-New Class generalization setting. Table 2 presents the performance comparison among the prompt tuning baselines, PromptKD, and the models augmented with our proposed RSM. Specifically, the RSM-augmented model demonstrates superior generalization performance on 6 out of 10 datasets, as well as in the average performance across all datasets under the 16-shot

Table 2. Comparison on Cross-datasets. Here, methods marked with † are trained under the 16-shot setting, while those marked with ‡ are trained under the full-shot setting.

| Method | Caltech | Pets | Cars | Flowers | Food | Aircraft | SUN | DTD | EuroSAT | UCF | Avg. |
|---|---|---|---|---|---|---|---|---|---|---|---|
| CLIP | 96.84 | 94.00 | 95.40 | 68.14 | 85.30 | 18.47 | 64.15 | 41.92 | 46.39 | 66.55 | 65.17 |
| CoOp | 93.70 | 89.14 | 64.51 | 68.71 | 85.30 | 18.47 | 64.15 | 41.92 | 46.39 | 66.55 | 63.88 |
| MaPLe | 93.53 | 90.49 | 65.57 | 72.23 | 86.20 | 24.74 | 67.01 | 46.49 | 48.06 | 68.69 | 66.30 |
| PromptSRC | 93.71 | 90.39 | 65.51 | 70.37 | 86.37 | 23.27 | 67.36 | 46.24 | 43.78 | 68.44 | 65.54 |
| PromptKD† | 93.40 | 75.03 | 71.25 | 73.04 | 84.63 | 24.84 | 67.50 | 47.34 | 38.35 | 72.39 | 64.78 |
| PromptKD‡ | 88.95 | 91.96 | 73.99 | 75.11 | 88.92 | 25.97 | 68.52 | 55.52 | 62.00 | 75.63 | 70.66 |
| PKD + RSM† | 93.74 | 87.96 | 70.84 | 73.94 | 84.73 | 24.46 | 66.77 | 52.52 | 39.75 | 72.27 | 66.70 |
| PKD + RSM‡ | 93.13 | 91.57 | 73.25 | 75.13 | 88.95 | 25.90 | 68.23 | 55.24 | 60.79 | 75.36 | 70.76 |

setting. Notably, under the 16-shot setting, RSM yields significant improvements in generalization performance when applied to PromptKD, with gains of 12.93% on OxfordPets, 5.18% on DTD, and 1.40% on EuroSAT. These improvements result in an average gain of 1.92% across all datasets. Clearly, this analysis consistently demonstrates that the proposed RSM module significantly improves the performance of vision-language unsupervised knowledge distillation models.

Note that, to further assess the effectiveness of RSM, we also integrate it into a simplified architecture (referred to as Simple), with the corresponding evaluation provided in the appendix.

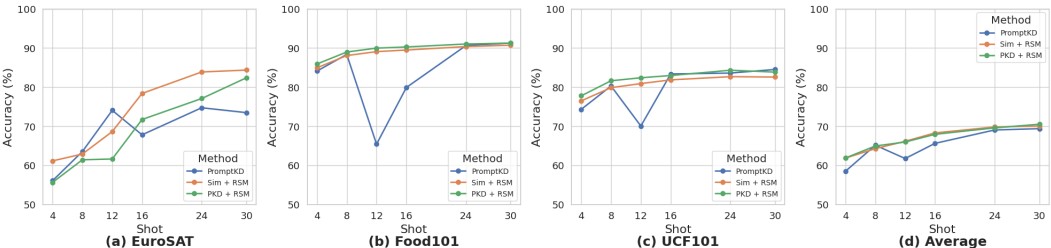

Figure 5. Performance comparison across three datasets and their averages under different shot.

### 4.2.3 EFFECT OF DIFFERENT SHOT SETTINGS ACROSS METHODS

Figure 5 illustrates the performance curves across different shot settings for three methods, PromptKD, Simple + RSM and PKD + RSM. Notably, the methods incorporating our proposed RSM exhibit consistent improvements as the number of training shots increases. Such results clearly indicate that RSM facilitates effective student–teacher knowledge transfer, even under low-data regimes.

## 5 CONCLUSION

Unsupervised Knowledge distillation has re-emerged as a common strategy for transferring knowledge from large pre-trained models to smaller ones across diverse downstream tasks. However, this paradigm exhibits limited generalization to samples with ambiguous or boundary-adjacent representations, often requiring large-scale training data to accurately delineate semantic boundaries in the representation space. Such limitations originate from the lack of directional guidance in the training objective, ultimately constraining the applicability of unsupervised knowledge distillation to multi-modal tasks. In this study, we propose **A**ligned **M**anifold **E**ntropy for Robust Unsupervised Vision-Language Distillation (AME), a method that enforces intra-class compression of cross-modal representations by applying entropy minimization over a reconfigured shared manifold. This mechanism enables the model to structurally align cross-modal representations during training, including those that are ambiguous or lie near decision boundaries, thereby enhancing training robustness and preserving generalization under data scarcity. Extensive experiments and theoretical analysis clearly indicate that AME effectively align cross-modal representation and achieve clearer separation, therefore facilitates the distillation process with both improved robustness and superior generalization performance. Crucially, this work proposes a lightweight, principled paradigm that paves the way for broader application of vision-language knowledge distillation in real-world tasks.

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
