## A  DISCLOSURE OF GENAI

Generative AI tools are utilized solely for polishing advice and grammar correction during the preparation of this paper. All content, such as research motivation, questions, experiments, and conclusions, is original contributions of the authors.

## B  PSEUDOCODE OF AME

Table 3: Algorithm 1  Pseudocode of AME in PyTorch.

| Definitions | Operations |
| --- | --- |
| # tea_t: text encoder of teacher CLIP | # init |
| # tea_i: image encoder of teacher CLIP | f_txt_t = tea_t(txt_of_all_classes) |
| # stu_i: image encoder of student CLIP | |
| # l_tea: teacher output logits | # forward |
| # l_stu: student output logits | for img in unlabeled_dataset: |
| # C: reconfigured shared manifold |   f_img_t = tea_i(img) |
| # Project Function: MLP |   f_img_s = stu_i(img) |
|   linear1: Linear |   f_txt_t* = MLP(f_txt_t) |
|   relu: QuickGELU() |   f_img_s* = ConvP(f_img_s) |
|   linear2: Linear |   C = concat(f_txt_t*, f_img_s*) |
| # Project Function: ConvP |   H(C) = $-\sum_i p(c_i) \log p(c_i)$ |
|   Conv2d1: Conv2d |   l_tea = f_img_t * f_txt_t.t() |
|   BatchNorm2d |   l_stu = f_img_s * f_txt_t.t() |
|   relu: QuickGELU() |   loss = KLDivergence(l_stu, l_tea) + H(C) |
|   Conv2d2: Conv2d |   loss.backward() |

## C  THEORETICAL ANALYSIS (CONTINUED)

### C.1  GENERALIZATION BOUND VIA ENTROPY REGULARIZATION.

**Corollary 2.** *By incorporating entropy minimization over the reconfigured shared manifold $M$, which encodes the projected features of the student model, into the distillation framework, the mutual information between the model parameters and the training data is implicitly constrained. This leads to the generalization bound:*

$$\mathbb{E}_{\mathcal{S},\theta}\big[L(\theta;\mathcal{D}) - L(\theta;\mathcal{S})\big] \;\lesssim\; \sqrt{\frac{2(\delta + \epsilon)}{n}}, \tag{10}$$

*where $\theta$ denotes the parameters of the student model, $\mathcal{D}$ and $\mathcal{S}$ represent the training and test datasets respectively, and $n$ is the number of training samples. The term $\delta$ quantifies the entropy $H(M)$ of the reconfigured shared manifold $M$, while $\epsilon$ captures residual dependencies that are not eliminated through the information entropy $H(M)$.*

This corollary demonstrates that the reconfigured shared manifold achieves an effective balance between compression and information preservation. Specifically, the entropy term $\delta = H(M)$ is explicitly minimized via the loss function, while the residual mutual information $\epsilon = I(\theta; \mathcal{S} \mid M)$ is implicitly suppressed, facilitated by the inclusion of a learnable projection function. In other words, the combined quantity $\delta + \epsilon$ is strictly smaller than the original mutual information $I(\theta; \mathcal{S})$ under the KL-only setting. This reduction tightens the generalization bound, thereby highlighting the effectiveness of entropy minimization within the reconfigured shared manifold.

## C.2 EFFECT OF SAMPLE SIZE.

As shown in Corollary 2, the generalization bound decreases at a rate inversely proportional to the square root of the sample size $n$. Formally, it can be expressed as:

$$\mathcal{O}\left(\sqrt{\frac{\delta+\epsilon}{n}}\right). \tag{1}$$

When the sample size $n$ is large, the generalization gap naturally diminishes, and the contribution of entropy regularization becomes marginal, as data sufficiency dominates the learning dynamics. In contrast, under low-data regimes, the entropy term $\mathcal{H}(M)$ becomes crucial. By minimizing $\delta$, it compensates for the effect of limited data and helps maintain a low generalization error. This observation highlights the distinctive advantage of entropy regularization in cross-modal knowledge distillation under low-data regimes.

# D PROOF

*Proof of Corollary 1.* Given the definition of conditional mutual information:

$$I(W; V \mid S) = H(W \mid S) + H(V \mid S) - H(W, V \mid S), \tag{11}$$

where $H(W \mid S)$ and $H(V \mid S)$ denote the conditional entropy of the text and image embeddings, respectively.

The KL divergence term $\text{KL}(P\|Q)$ minimizes the discrepancy between distributions $P$ and $Q$, thereby increasing $I(W; V \mid S)$ by aligning the probabilistic outputs of the teacher $(T)$ and student $(V)$. In addition, the entropy term $H(M)$ serves as an explicit regularizer on the shared manifold. Since $H(M)$ approximates $H(W, V \mid S)$, minimizing $H(M)$ effectively reduces $H(W, V \mid S)$ and thus increases the conditional mutual information, strengthening the dependency between text and image embeddings, as follows:

$$I(W; V \mid S) = H(W \mid S) + H(V \mid S) - H(W, V \mid S) \downarrow. \tag{12}$$

Here, since the total loss mainly targets the joint alignment between $T$ and $V$, rather than modifying the marginal distributions, we assume that $H(W \mid S)$ and $H(V \mid S)$ remain approximately constant. □

*Proof of Corollary 2.* Given the setting of theorem, assume:

- The distillation KL loss $\mathcal{L}_{\text{KD}}$ increases the dependency between model parameters $\theta$ and the training data $\mathcal{S}$, such that the mutual information is defined as $I(\theta; \mathcal{S})$.

- The entropy regularization term $\omega H(M)$ constrains the expressiveness of the reconfigured latent space $M$.

Since $M$ is a deterministic function of $(\theta, \mathcal{S})$, the data processing inequality gives:

$$I(M; \mathcal{S}) \leq H(M). \tag{13}$$

By applying the chain rule of mutual information, we obtain:

$$I(\theta; \mathcal{S}) = I(M; \mathcal{S}) + I(\theta; \mathcal{S} \mid M) \leq H(M) + \epsilon \leq \delta + \epsilon, \tag{14}$$

where $\epsilon = I(\theta; \mathcal{S}) - I(M; \mathcal{S})$ quantifies the residual dependency not captured by $M$. Therefore, the generalization error bound becomes:

$$\mathbb{E}_{\mathcal{S},\theta}\big[L(\theta; \mathcal{D}) - L(\theta; \mathcal{S})\big] \leq \sqrt{\frac{2(\delta + \epsilon)}{n}}. \tag{15}$$

□

## D.1 DATASETS.

To ensure consistency with prior works (Zhou et al., 2022b;a; Khattak et al., 2023a; Gao et al., 2021a; Li et al., 2024), we adopt the evaluation protocol on the same datasets. For the Base-to-New Class Generalization, eleven image datasets are used. In the cross-dataset generalization setting, we follow the protocol of PromptKD, evaluating on ten datasets with ImageNet excluded. Specifically, Caltech101 (Fei-Fei et al., 2004) and ImageNet (Deng et al., 2009) are categorized as generic-object datasets; FGVCAircraft (Maji et al., 2013), Flowers102 (Nilsback & Zisserman, 2008), Food101 (Bossard et al., 2014), OxfordPets (Parkhi et al., 2012) and StanfordCars (Krause et al., 2013) are fine-grained image datasets; DTD (Cimpoi et al., 2013) and EuroSAT (Helber et al., 2019) are texture and satellite datasets, respectively, while SUN397 (Xiao et al., 2010) and UCF101 (Soomro et al., 2012) are used for scene and action recognition tasks, respectively.

# E    EXTENDED RESULTS ON BASE-TO-NEW AND CROSS-DATASET GENERALIZATION

In this appendix, we extend the experiments on Base-to-New class generalization and Cross-Dataset generalization reported in the main text. Specifically, we supplement the analysis by evaluating additional backbone models equipped with our proposed module. These results further verify the robustness and adaptability of the paradigm across diverse architectures, demonstrating consistent improvements over the corresponding baselines.

Table 4. Comparison with existing approaches across 11 datasets and their averages. Here, RSM modules are appended to Simple distillation architectures, and are denoted by "+ RSM". While † indicates 16-shot training and ‡ indicates full-shot training, with the number of shots shown in parentheses after the dataset name.

### (a) Average

| Name | Base | New | HM |
|---|---|---|---|
| CLIP | 69.34 | 74.22 | 71.70 |
| CoOp | 82.69 | 63.22 | 71.66 |
| MaPLe | 82.28 | 75.14 | 78.55 |
| PromptSRC | 84.19 | 75.28 | 79.49 |
| PromptKD† | 80.36 | 74.64 | 77.40 |
| PromptKD‡ | 85.21 | 79.13 | 82.06 |
| Sim + RSM† | 82.57 | 77.18 | 79.78 |
| Sim + RSM‡ | 85.08 | 79.21 | 82.04 |

### (b) ImageNet (≈ 1281)

| Name | Base | New | HM |
|---|---|---|---|
| CLIP | 72.43 | 68.14 | 70.22 |
| CoOp | 76.47 | 67.88 | 71.92 |
| MaPLe | 76.66 | 70.54 | 73.47 |
| PromptSRC | 77.36 | 70.58 | 73.81 |
| PromptKD† | 77.02 | 71.03 | 73.90 |
| PromptKD‡ | 76.04 | 70.74 | 73.30 |
| Sim + RSM† | 74.53 | 68.84 | 71.57 |
| Sim + RSM‡ | 74.14 | 70.11 | 72.07 |

### (c) Caltech101 (≈ 41)

| Name | Base | New | HM |
|---|---|---|---|
| CLIP | 96.84 | 94.00 | 95.40 |
| CoOp | 98.00 | 89.81 | 93.73 |
| MaPLe | 97.74 | 94.36 | 96.02 |
| PromptSRC | 97.98 | 93.92 | 95.91 |
| PromptKD† | 98.34 | 95.23 | 96.76 |
| PromptKD‡ | 98.90 | 96.47 | 97.67 |
| Sim + RSM† | 97.89 | 95.31 | 96.58 |
| Sim + RSM‡ | 98.21 | 96.07 | 97.13 |

### (d) OxfordPets (≈ 79)

| Name | Base | New | HM |
|---|---|---|---|
| CLIP | 91.17 | 97.26 | 94.12 |
| CoOp | 93.67 | 95.29 | 94.47 |
| MaPLe | 95.43 | 97.76 | 96.58 |
| PromptSRC | 95.43 | 97.30 | 96.35 |
| PromptKD† | 90.13 | 92.54 | 91.32 |
| PromptKD‡ | 93.39 | 95.79 | 94.57 |
| Sim + RSM† | 93.34 | 97.13 | 95.20 |
| Sim + RSM‡ | 95.46 | 98.17 | 96.80 |

### (e) StanfordCars (≈ 32)

| Name | Base | New | HM |
|---|---|---|---|
| CLIP | 63.37 | 74.89 | 68.65 |
| CoOp | 78.12 | 60.40 | 68.13 |
| MaPLe | 72.94 | 74.00 | 73.47 |
| PromptSRC | 78.53 | 75.23 | 76.84 |
| PromptKD† | 79.48 | 81.40 | 80.43 |
| PromptKD‡ | 76.75 | 78.21 | 77.47 |
| Sim + RSM† | 78.30 | 79.79 | 79.04 |
| Sim + RSM‡ | 80.22 | 81.37 | 80.79 |

### (f) Flowers102 (≈ 40)

| Name | Base | New | HM |
|---|---|---|---|
| CLIP | 72.08 | 77.80 | 74.83 |
| CoOp | 97.60 | 59.67 | 74.06 |
| MaPLe | 95.92 | 72.46 | 82.56 |
| PromptSRC | 98.04 | 74.82 | 84.87 |
| PromptKD† | 95.38 | 77.28 | 85.38 |
| PromptKD‡ | 96.26 | 80.05 | 87.41 |
| Sim + RSM† | 98.10 | 81.18 | 88.84 |
| Sim + RSM‡ | 99.02 | 81.87 | 89.63 |

### (g) Food101 (= 500)

| Name | Base | New | HM |
|---|---|---|---|
| CLIP | 90.10 | 91.22 | 90.66 |
| CoOp | 88.33 | 82.26 | 85.19 |
| MaPLe | 90.71 | 92.05 | 91.38 |
| PromptSRC | 90.84 | 74.82 | 84.87 |
| PromptKD† | 78.56 | 81.29 | 79.90 |
| PromptKD‡ | 92.39 | 93.78 | 93.08 |
| Sim + RSM† | 88.48 | 90.51 | 89.48 |
| Sim + RSM‡ | 92.16 | 93.39 | 92.77 |

### (h) FGVCAircraft (≈ 33)

| Name | Base | New | HM |
|---|---|---|---|
| CLIP | 27.19 | 36.29 | 31.09 |
| CoOp | 40.44 | 22.30 | 28.75 |
| MaPLe | 37.44 | 35.61 | 36.50 |
| PromptSRC | 42.82 | 36.77 | 39.57 |
| PromptKD† | 44.42 | 39.81 | 41.99 |
| PromptKD‡ | 47.84 | 41.33 | 44.35 |
| Sim + RSM† | 43.68 | 37.27 | 40.22 |
| Sim + RSM‡ | 45.44 | 38.77 | 41.84 |

### (i) SUN397 (≈ 40)

| Name | Base | New | HM |
|---|---|---|---|
| CLIP | 69.36 | 75.35 | 72.23 |
| CoOp | 80.60 | 65.89 | 72.51 |
| MaPLe | 80.82 | 78.70 | 79.75 |
| PromptSRC | 82.59 | 78.71 | 80.60 |
| PromptKD† | 82.64 | 80.48 | 81.55 |
| PromptKD‡ | 83.48 | 81.38 | 82.42 |
| Sim + RSM† | 81.48 | 79.50 | 80.48 |
| Sim + RSM‡ | 82.81 | 80.84 | 81.81 |

### (j) DTD (= 60)

| Name | Base | New | HM |
|---|---|---|---|
| CLIP | 53.24 | 59.90 | 56.37 |
| CoOp | 79.44 | 41.18 | 54.24 |
| MaPLe | 80.36 | 59.18 | 68.16 |
| PromptSRC | 82.83 | 61.19 | 70.39 |
| PromptKD† | 75.42 | 60.51 | 67.15 |
| PromptKD‡ | 85.42 | 71.05 | 77.58 |
| Sim + RSM† | 81.94 | 68.16 | 74.42 |
| Sim + RSM‡ | 84.34 | 70.49 | 76.80 |

### (k) EuroSAT (≈ 1350)

| Name | Base | New | HM |
|---|---|---|---|
| CLIP | 56.48 | 64.05 | 60.03 |
| CoOp | 92.19 | 54.74 | 68.69 |
| MaPLe | 94.07 | 73.23 | 82.35 |
| PromptSRC | 93.35 | 69.29 | 79.54 |
| PromptKD† | 75.47 | 61.55 | 67.80 |
| PromptKD‡ | 97.32 | 80.27 | 87.98 |
| Sim + RSM† | 85.41 | 72.41 | 78.38 |
| Sim + RSM‡ | 96.58 | 79.92 | 87.47 |

### (l) UCF101 (≈ 75)

| Name | Base | New | HM |
|---|---|---|---|
| CLIP | 70.53 | 77.50 | 73.85 |
| CoOp | 84.69 | 56.05 | 67.46 |
| MaPLe | 83.00 | 78.66 | 80.77 |
| PromptSRC | 86.35 | 78.71 | 82.35 |
| PromptKD† | 87.12 | 79.90 | 83.36 |
| PromptKD‡ | 89.50 | 81.32 | 85.22 |
| Sim + RSM† | 85.07 | 78.85 | 81.85 |
| Sim + RSM‡ | 87.50 | 80.31 | 83.75 |

For Base-to-New Class Generalization, for the average performance across 11 datasets, by integrating our proposed module, Simple + RSM achieves competitive results of 82.57%, 77.18% and 79.78%, Notably, it can be clearly seen that Simple + RSM surpass the best generalization provide by baselines on both the New class and HM metrics, and achieve the best HM on 7 out of 11 datasets,

Table 5. Comparison on Cross-datasets. Here, RSM is appended to a simple distillation model. Methods marked with † are trained under the 16-shot setting, while those marked with ‡ are trained under the full-shot setting.

| Method | Caltech | Pets | Cars | Flowers | Food | Aircraft | SUN | DTD | EuroSAT | UCF | Avg. |
|---|---|---|---|---|---|---|---|---|---|---|---|
| CLIP | 96.84 | 94.00 | 95.40 | 68.14 | 85.30 | 18.47 | 64.15 | 41.92 | 46.39 | 66.55 | 65.17 |
| CoOp | 93.70 | 89.14 | 64.51 | 68.71 | 85.30 | 18.47 | 64.15 | 41.92 | 46.39 | 66.55 | 63.88 |
| MaPLe | 93.53 | 90.49 | 65.57 | 72.23 | 86.20 | 24.74 | 67.01 | 46.49 | 48.06 | 68.69 | 66.30 |
| PromptSRC | 93.71 | 90.39 | 65.51 | 70.37 | 86.37 | 23.27 | 67.36 | 46.24 | 43.78 | 68.44 | 65.54 |
| PromptKD† | 93.40 | 75.03 | 71.25 | 73.04 | 84.63 | 24.84 | 67.50 | 47.34 | 38.35 | 72.39 | 64.78 |
| PromptKD‡ | 88.95 | 91.96 | 73.99 | 75.11 | 88.92 | 25.97 | 68.52 | 55.52 | 62.00 | 75.63 | 70.66 |
| Sim + RSM† | 93.17 | 87.31 | 69.68 | 73.27 | 83.86 | 24.26 | 66.00 | 50.95 | 48.72 | 71.42 | 66.86 |
| Sim + RSM‡ | 88.71 | 85.30 | 61.82 | 70.04 | 85.63 | 20.65 | 62.72 | 49.90 | 52.42 | 69.19 | 64.64 |

including Caltech101, StanfordCars, Flowers102, FGVCAircraft, SUN397 DTD and UCF101. With the full-shot training setting, RSM significantly improves the performance of Simple-based distillation models across all metrics on OxfordPets, StanfordCars, and Flowers102.

For Cross-Dataset Generalization, the RSM-augmented model demonstrates superior generalization performance on 6 out of 10 datasets, as well as in the average performance across all datasets under the 16-shot setting.

# F  FURTHER EXPERIMENTS (SUPPLEMENTARY)

To further investigate the effectiveness and adaptability of our proposed RSM module, we conduct a series of extended analyses. Specifically, we integrate RSM into two well-established prompt tuning approaches, MaPLe and PromptSRC, to assess its compatibility across different architectures. In addition, we evaluate its performance under different shot settings by comparing it with the original PromptKD. We also analyze the role of the learnable projection used in the reconfigured shared manifold. These investigations are presented and discussed in the following sections.

Table 6: Effectiveness of RSM for prompt tuning on the DTD.

| Method | Base | New | HM |
|---|---|---|---|
| MaPLe | 81.75 | 53.50 | 64.67 |
| MaPLe + RSM | 82.29 | 55.56 | 66.33 |
| Δ | +0.54 | +2.06 | +1.66 |
| PromptSRC | 82.83 | 61.19 | 70.39 |
| PromptSRC + RSM | 82.48 | 63.41 | 71.70 |
| Δ | -0.35 | +2.22 | +1.31 |

## F.1  EFFECTIVENESS OF RSM FOR THE PROMPT TUNING PARADIGM

Having evaluated the generalization performance of the prompt tuning approaches augmented with our proposed RSM, Table 6 represents an analysis for adaptability on the prompt tuning paradigm under supervised training manner. In specific, our method significantly improves the performance of MaPLe, yielding gains of 0.54%, 2.06%, and 1.66% on Base, New, and HM, respectively. For PromptSRC, improvements of 2.22% and 1.31% can be also observed on New and HM. These findings suggest that the proposed RSM is broadly applicable across diverse vision-language knowledge distillation model frameworks and consistently contributes to improved generalization performance on a wide range of downstream tasks.

## F.2 EFFECT ON LEARNABLE PROJECTION

Table 7: Effect of the learnable projection on performance. Here, B and N stand for Base and New class, respectively.

| Learn. Proj. | DTD | | | EuroSAT | | | Average | | |
|---|---|---|---|---|---|---|---|---|---|
| | B | N | HM | B | N | HM | B | N | HM |
| | 83.76 | 67.43 | 74.71 | 74.79 | 60.68 | 67.00 | 82.84 | 76.75 | 79.68 |
| ✓ | 84.15 | 69.16 | 75.92 | 79.94 | 65.03 | 71.72 | 83.31 | 77.30 | 80.19 |
| Δ | +0.39 | +1.73 | +1.21 | +5.15 | +4.35 | +4.72 | +0.46 | +0.54 | +0.51 |

Table 7 evaluates the effectiveness of the learnable projection used in the reconfigured latent space across two representative datasets, along with the average performance over all 11 datasets. For example, the average Base, New, and HM scores improve from 82.84%, 76.75% and 79.68% to 83.31%, 77.30% and 80.19%, respectively. Note that a substantial improvement is observed on EuroSAT, with significant gains of 5.15%, 4.35% and 4.72% for Base, New, and HM, respectively.

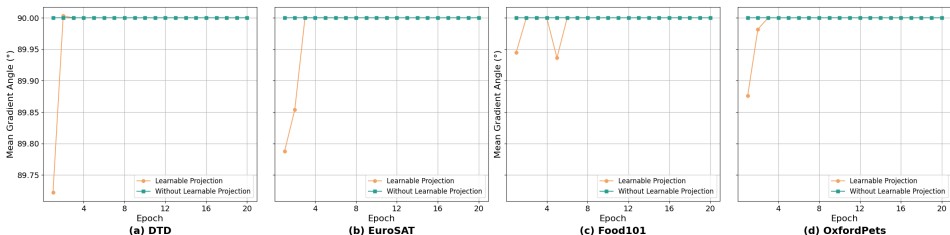

Figure 6. Gradient angle between the KL loss and the infomation entropy minimization loss.

Notably, Figure 6 demonstrate that information entropy minimization begins to interact with the training dynamics when a learnable prompt is introduced, as evidenced by the gradient angle between the KL loss and the entropy loss dropping below 90 degrees across four datasets, including DTD, EuroSAT, Food101, and OxfordPets. In contrast, without the learnable projection, the gradient angle stays around 90 degrees, indicating orthogonality and weak interaction between the two training objectives. This finding aligns with our motivation to project multi-modal features into a shared manifold, thereby enabling effective feature compression and enhancing intra-class determinacy.

## F.3 CLASS–FEATURE RELEVANCE HEATMAPS (ALL DATASETS)

Similar to the observation in Figure 3, this figure 7 corroborates that our paradigm yields stronger and more pervasive activation patterns across dimensions and classes, thereby aligning a larger fraction of the representation space with class-discriminative signals.

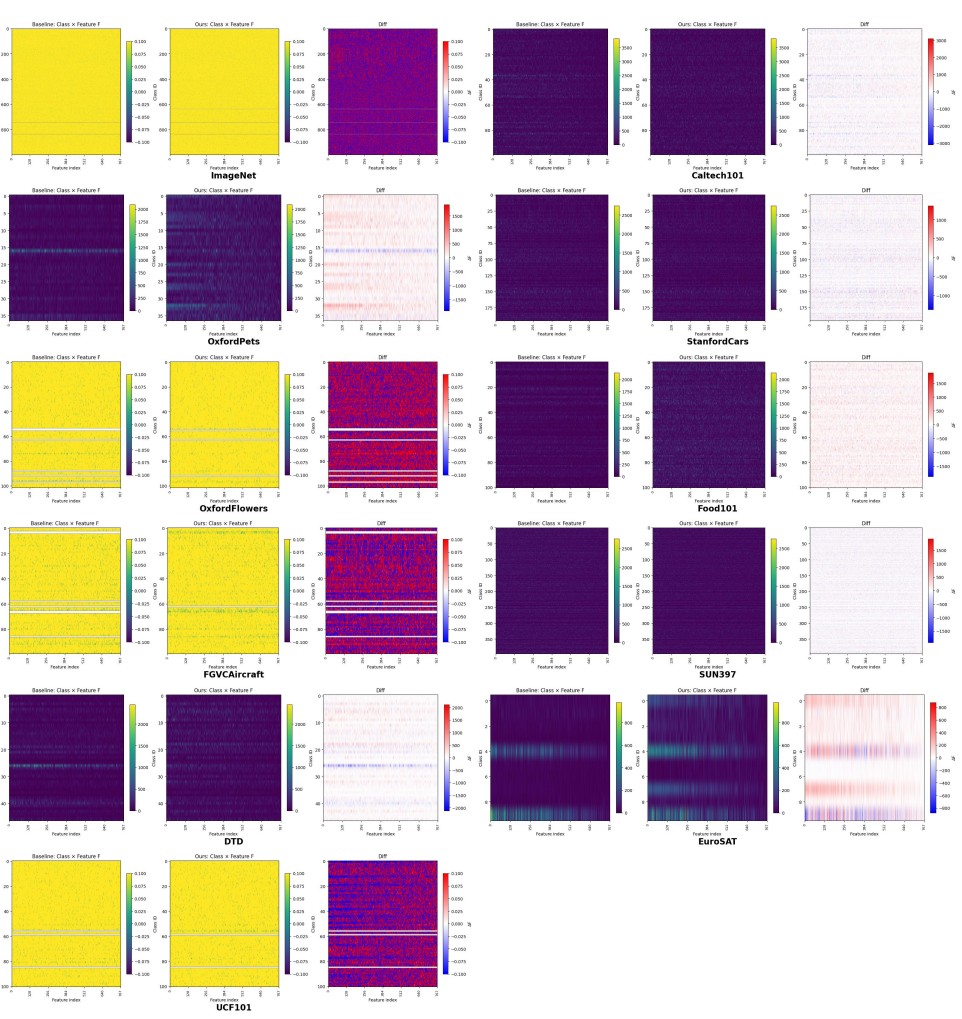

Figure 7. Class–feature Relevance Heatmaps over all 11 datasets. This figure illustrates the class–feature relevance heatmaps computed via one-vs-rest F-scores across representation dimensions, where each column corresponds to a dimension and each row to a class. Brighter intensities signify stronger discriminative utility of a dimension for separating the target class from all others.