# OpenReview forum: "AME: ALIGNED MANIFOLD ENTROPY FOR ROBUST UNSUPERVISED VISION-LANGUAGE DISTILLATION"
_ICLR.cc/2026/Conference — ICLR 2026 Conference Withdrawn Submission_

### Official Review · Reviewer_HdQh · 2025-10-29

**Soundness:** 3
**Presentation:** 3
**Contribution:** 2
**Rating:** 2
**Confidence:** 5

**Summary:**

This paper introduces Aligned Manifold Entropy (AME), a framework for unsupervised vision-language distillation. The core idea is to apply entropy minimization over a Reconfigured Shared Manifold (RSM) where textual and visual embeddings are jointly projected using lightweight mappings (MLP for text, convolution for image). The authors argue that conventional KL-based distillation lacks directional alignment between modalities and propose AME to restore this property. A theoretical analysis -- based on mutual information and entropy regularization -- is presented to justify improved generalization. Empirically, AME is evaluated on 11 datasets and shows consistent but small gains compared to PromptKD and related baselines, especially under few-shot or low-data regimes.

**Strengths:**

1. The paper identifies a persistent issue in multi-modal distillation -- directionless scalar objectives -- and connects it with cross-modal misalignment.
2. The authors provide a high-level conceptual link between entropy regularization and representational alignment, making the method intuitively appealing.
3. Empirical results cover multiple datasets and tasks, showing consistent if modest improvement in both base-to-new and cross-dataset generalization settings.
4. The proposed RSM module is lightweight and easy to integrate into existing frameworks.

**Weaknesses:**

1. The main novelty -- adding an entropy regularization term to a reconfigured feature manifold -- is conceptually modest.
Formally, the proposed loss $L_{total}$=$L_{kd}+\lambda H(M)$ resembles standard entropy minimization used in semi-supervised and domain adaptation literature. The “aligned manifold” terminology does not correspond to a fundamentally new algorithmic idea; the projection via simple linear/MLP layers followed by entropy compression is functionally equivalent to known low-entropy regularizers. The contribution thus appears incremental.

2. The theoretical sections rephrase common results from information theory without adding substantive rigor. The assumption that minimizing entropy $H(M)$ increases conditional mutual information $I(W;V∣S)$ is asserted but not proven; it requires prerequisites like invertibility of projections that are neither justified nor realistic. The “generalization bound” uses undefined constants $\delta$, $\epsilon$ and postulates a reduction without formal derivation or empirical verification. Consequently, the theoretical framework lacks robustness and provides little provable insight beyond intuition.

3. The practical RSM implementation -- concatenating feature projections and computing global entropy -- does not convincingly realize the theoretical narrative of manifold alignment. No quantitative measure (e.g., alignment scores, manifold curvature, inter/intra-class distance) supports the claim that this projection induces geometric consistency. The empirical behavior could simply result from regularization effects rather than true “directional alignment.”

4. Experimental improvements are modest (≈1–3%) and sometimes within the variance of training noise. There is no demonstration that reduced entropy correlates with improved cross-modal consistency or generalization. Visualizations (t-SNE, feature heatmaps) are qualitative and not backed by statistical analysis. The experiments are limited to standard classification tasks; no evaluation on alignment-sensitive tasks (e.g., retrieval, captioning) is provided to substantiate the claimed advantages. Overall, empirical validation is too shallow to confirm the theoretical claims.

5. The objective combines entropy minimization with distillation. This combination might suppress modality-specific variability and harm fine-grained discrimination. The paper does not explore this potential trade-off or provide ablations isolating the role of each component.

**Questions:**

See weaknesses.

Additional Questions:
1. The paper claims that minimizing the manifold entropy $H(M)$ increases the conditional mutual information $I(W;V∣S)$, but the proof appears qualitative. Could the authors provide a more formal derivation or an empirical estimation of $I(W;V∣S)$ to support this argument?
2. The proposed “directional alignment” effect of entropy minimization is mostly illustrated through qualitative visualizations. Can the authors report a quantitative metric, such as cosine similarity alignment scores, inter/intra-class variance, or canonical correlation to substantiate this claim?
3. The reconfigured shared manifold $M$ is created via simple projection functions before entropy minimization. How can the authors ensure that such a projection does not become a trivial bottleneck or induce over-compression that harms class diversity?
4. The theoretical generalization bound in Corollary 2 defines $\delta=H(M)$ and $\epsilon=I(\theta; S∣M)$. Have the authors measured or tracked these quantities during training to empirically validate the claimed link between entropy compression and generalization improvement?

---

### Official Review · Reviewer_o3zJ · 2025-11-01

**Soundness:** 2
**Presentation:** 2
**Contribution:** 3
**Rating:** 4
**Confidence:** 3

**Summary:**

The paper proposes Aligned Manifold Entropy (AME) for unsupervised vision–language distillation. Given a pretrained teacher and a lightweight student, AME first projects text and image embeddings into a low‑dimensional Reconfigured Shared Manifold (RSM), then computes per‑row scalar scores ($s_t$) by feature‑wise averaging, converts them to probabilities ($p_t=\mathrm{softmax}(s)_t$), and minimizes the entropy $(H(M)=-\sum_t p_t\log p_t\)$. The total loss is

$L_{\text{Total}}=L_{\text{KD}} + \omega H(M)$,

where $L_{\text{KD}}=\tau^2\,D_{\mathrm{KL}}\big(\sigma(z_t/\tau)\,\Vert\,\sigma(z_s/\tau)\big)$.

The theoretical section argues that combining KD with entropy compression yields class‑wise prototype formation and inter‑class margins in RSM.

Empirically, the method reports modest improvements over PromptKD on Base→New

**Strengths:**

- AME is a clean, lightweight add-on that merges information-theoretic entropy minimization with KD in a shared, low-dimensional manifold. The formalization of RSM—"row-wise concatenation of projected text and image embeddings", averaging to $s_t$, softmax to $p_t$, and entropy $H(M)$—is an explicit design that could be plugged into other UKD pipelines.
- The paper motivates that KD’s logit-only supervision may lack directionality in multi-modal alignment and presents AME as an information-theoretic complement. The figures (Heatmaps, t-SNE) visually support the claimed structural effects.
- AME is backbone-agnostic and "plug-and-play," which is attractive for practitioners who want a minimal change to distillation pipelines.

**Weaknesses:**

(W1) Equation (7) is incorrect / inconsistent.
The paper wants the gradient "with respect to $s_t$", but the equation instead shows $\partial H/\partial p_t$ and includes a redundant summation:

$\text{In paper:}\quad \frac{\partial H}{\partial p_t} = -\sum_t (1+\log p_t)$

$
\frac{\partial H}{\partial p_k}=-(1+\log p_k),\qquad
\frac{\partial H}{\partial s_t} = \sum_i \frac{\partial H}{\partial p_i}\frac{\partial p_i}{\partial s_t}.
$

It might be clearer to revise Eq. (7) and specify whether the derivative is taken with respect to $s_t$ or $p_t$, s the current form could make the subsequent discussion on gradient flow and collapse somewhat ambiguous.

(W2) Theorem 1 margin term $\zeta$ is used but not defined.
The theorem concludes $\|\mathbf h_c-\mathbf h_{c'}\|\ge \zeta>0$, but there is no explicit definition of $\zeta$ (norm, dependence on model/data, conditions under which $\zeta$ exists).

(W3) Limited empirical dominance; small average gains and mixed per-dataset results.
– Averages: 16-shot HM: 77.40 → 80.19 (+2.79); full-shot HM: 82.06 → 82.96 (+0.90). Gains are real but not large.
– Mixed results: e.g., ImageNet 16-shot HM drops 73.90 → 73.00; SUN397 16-shot HM roughly ties 81.55 ↔ 81.08; UCF101 full-shot HM drops 85.22 → 83.58.

(W4) No repeatability statistics in tables.
Authors report 3 seeds averaged but give no standard deviations / confidence intervals.

(W5) The weighting $\omega$ is critical yet lacks ablation / sensitivity and rationale.
$\omega$ controls the trade-off between KD and entropy compression. Authors fix $\omega=50$, but provide no ablation (e.g., $\omega\in\{0,1,5,10,25,50,100\}$) nor a justification for 50. This is important because large $\omega$ should push $p$ toward one-hot (collapse) while small $\omega$ reduces AME’s effect.

**Questions:**

Please provide responses to the points mentioned in the Weaknesses section.

---

### Official Review · Reviewer_s7ZV · 2025-11-01

**Soundness:** 2
**Presentation:** 2
**Contribution:** 2
**Rating:** 2
**Confidence:** 4

**Summary:**

This paper addresses a problem in applying Unsupervised Knowledge Distillation (KD) to Vision-Language Models (VLMs) like CLIP. The authors note that CLIP's directional training losses conflict with standard, directionless KD objectives like KL divergence. To resolve this, the paper proposes Aligned Manifold Entropy (AME), which aims to make the student model's image and text representations more compact. This is achieved by minimizing entropy on a Reconfigured Shared Manifold (RSM).

**Strengths:**

The paper identifies an interesting and fundamental discrepancy in typical Unsupervised KD applications for CLIP-style models.

**Weaknesses:**

**Lack of Motivation:** The proposed AME loss formulation lacks motivation. The paper explains how the loss is computed (e.g., Eq. 4, Eq. 5) but lacks a clear justification for why this specific formulation is chosen.

**Loss Formulation Concerns:**
- The loss in Eq (4) does not appear to account for varying scales across different dimensions, which could disproportionately weight certain dimensions.
- The intuition for minimizing entropy in Eq. (5) is unclear. This objective appears to favor a degenerate distribution (e.g., one highly positive row, with others negative before softmax), and it is not clear why this would improve representation alignment.

**Experimental Details Missing:** The experimental details are insufficient. For example, the training dataset used in section 4.2.2 is not specified.

**Missing Comparisons & Baselines:**
- Key comparisons to recent and relevant CLIP distillation methods are missing from the discussion and results [1]-[5].
- The evaluation is incomplete. Standard benchmarks, such as Image Classification in the Wild [6], are missing for Table 2.
- There are no retrieval results on standard datasets like MS-COCO and Flickr-30k.

**Results Analysis:** Table 2 suggests that the full-shot setting exhibits diminishing returns compared to the few-shot setting. Further analysis with fewer shots would be insightful.

**Clarity:** The writing is generally difficult to follow,.

[1] Yang, C., et al. (2024). Clip-kd: An empirical study of clip model distillation. In CVPR.

[2] Vasu, P. K. A., et al. (2024). Mobileclip: Fast image-text models through multi-modal reinforced training. In CVPR.

[3] Saidutta, Y. M., et al. (2024). CIFD: Controlled Information Flow to Enhance Knowledge Distillation. In NeurIPS.

[4] Sameni, S., et al. (2024). Building vision-language models on solid foundations with masked distillation. In CVPR.

[5] Wu, Kan, et al. "Tinyclip: Clip distillation via affinity mimicking and weight inheritance." Proceedings of the IEEE/CVF International Conference on Computer Vision. 2023.

[6] Li, C., et al. (2022). Elevater: A benchmark and toolkit for evaluating language-augmented visual models. In NeurIPS.

**Questions:**

See above.

---

### Official Review · Reviewer_3PZV · 2025-11-01

**Soundness:** 3
**Presentation:** 2
**Contribution:** 2
**Rating:** 4
**Confidence:** 3

**Summary:**

This paper introduces Aligned Manifold Entropy (AME), a framework for robust unsupervised vision–language distillation. Unlike traditional unsupervised knowledge distillation, which relies on scalar and directionless probability distributions, the authors address cross-modal misalignment from a distillation perspective by performing entropy compression on a reconfigured shared manifold (RSM), where image and text embeddings are jointly projected into a compact, directionally aligned space. Without modifying the original backbone, AME enforces geometric consistency between modalities, improving representation convergence and generalization in low-data regimes. Extensive experiments and theoretical analyses show that AME consistently enhances unsupervised VLM distillation, achieving superior performance across 11 datasets.

**Strengths:**

- The proposed AME framework (via the RSM module) can be seamlessly integrated into various existing distillation backbones (e.g., PromptKD, SimpleKD) without altering the underlying architecture. This plug-and-play nature enhances its practicality and compatibility with prior knowledge distillation methods

- Good finding on manifold misalignment within unsupervised vision–language knowledge distillation. The authors identify that conventional unsupervised KD prior works fail to preserve the directional geometry of multimodal embeddings. By introducing a shared manifold with entropy compression, the paper provides a principled solution to restore cross-modal alignment and improve generalization in unsupervised distillation settings.

**Weaknesses:**

- The performance gains of AME are mainly observed under 16-shot or cross-dataset conditions, while the improvement diminishes to less than 1% in full-data settings. This indicates that as the number of training samples increases, the benefit of geometric alignment through the Reconfigured Shared Manifold (RSM) becomes less pronounced. The proposed entropy-based regularization seems primarily effective in stabilizing representation geometry under data-scarce regimes, but its impact weakens when the data manifold is sufficiently covered by abundant supervision

- Insufficient benchmarking against recent unsupervised KD methods. The paper mainly compares AME with PromptKD but lacks broader evaluation against other recent unsupervised or label-free distillation approaches (e.g., KDPL 2024, COSMOS 2025, SILC 2024). Without such head-to-head baselines, it remains unclear whether the proposed manifold-based alignment truly provides stronger or more consistent benefits than other unsupervised objectives or prompt-based designs.

**Questions:**

Could the authors include comparisons with more recent unsupervised or label-free knowledge distillation frameworks  to clarify whether the proposed manifold-based alignment truly offers superior or complementary benefits over other prompt-free or self-distillation objectives?

---

### Note · Authors · 2025-11-25

I have read and agree with the venue's withdrawal policy on behalf of myself and my co-authors.